# Knowledge Collaboration among Tax Professionals through the Lens of a Community of Practice

Nurhidayah Bahar [1,*], Shamshul Bahri [2] and Zarina Zakaria [3]

1 Center for Software Technology and Management, Faculty of Information Science & Technology, The National University of Malaysia, Bangi 43600, Malaysia
2 Department of Management, Faculty of Business & Economics, University of Malaya, Kuala Lumpur 50603, Malaysia
3 Department of Accounting, Faculty of Business & Economics, University of Malaya, Kuala Lumpur 50603, Malaysia
* Correspondence: nbahar@ukm.edu.my; Tel.: +60-3-8921-6752

**Abstract:** This paper presents knowledge collaboration among tax professionals in a tax-knowledge context within Malaysian accounting associations through the conceptual lens of a community of practice. Semi-structured in-depth interviews were conducted with a total of 29 tax professionals. Additionally, data were also gathered from field notes and archival data. The findings revealed that the Malaysian accounting-professional associations reflected a community of practice. Knowledge collaboration occurs among members in this community in order to attain the highest standard of technical and professional competency in tax knowledge and practice. The findings from this study complement and expand previous research on CoP, knowledge management, and collaboration. The findings suggest exploring a better strategy to implement a central repository of knowledge acquired or generated by the members within the community to support the learning lifecycle.

**Keywords:** knowledge collaboration; community of practice; knowledge management; taxation; tax knowledge; tax system

## 1. Introduction

Collaboration, in general, offers several benefits, including a base to generate new knowledge, theories, and ideas, enhance subsequent advancement and growth, achieve competitive advantages, increase innovation and creativity, and improve members' motivation (Tan and Md. Noor 2013; Howells et al. 2012; Faraj et al. 2011). Ultimately, the relevant knowledge generated and shared during collaboration could be useful for personal and professional progression.

Knowledge collaboration (KC) can be regarded as the action of sharing, transferring, accumulating, transforming, and co-creating knowledge (Faraj et al. 2011). As communities are an important source of knowledge (Adler 2001), KC often occurs in a community of practice (CoP). In recent years, academics and those who work in the field of taxation (e.g., tax lawyers, tax consultants, tax agents) have demonstrated a significant interest in KC in CoPs. It is believed that CoPs could help foster collaboration among professionals, including doctors, lawyers, architects, consultants, and accountants (Jennings Mabery et al. 2013).

CoP often refers to a group of people who are formally or informally bound by shared competence and interests in a specific practice (Choi 2006); however, the way they operate differs according to their professions (Davenport 2001). For example, a CoP for healthcare professionals may operate differently from that of human resources professionals and academicians, and starkly differently from that for businessmen and industrialists (Nagy and Burch 2009). Consequently, CoPs have different goals, roles, and methods of communication (Kimmerle et al. 2013).

There are three important components in a CoP. The first is a domain in which members share the same or similar competence, the second is a community where members engage with each other, and the third is practice where they share related experiences (Venkatraman and Venkatraman 2018). The tax community, for example, can be considered as a CoP. Throughout the year, they produce seminars, training, and publications to share their experiences and know-how among themselves and the general public. These components, along with various artefacts, such as a virtual environment where Web 2.0 technologies are employed, help reify their foundation, tacit knowledge, and experiences, and ensure their survival (Zimitat 2007; Lara et al. 2017).

KC is important among members of a CoP. It is particularly popular in business studies and helps improve the efficiency of knowledge flow and exchange. Ultimately, this leads to the generation of value-added knowledge among members of either single-organization communities or multi-communities and organizations. Recently, KC has also gained popularity in CoP-related studies. This includes virtual CoPs and hybrid-virtual communities (Jarvenpaa and Majchrzak 2010; Faraj et al. 2011; Grabher and Ibert 2014) that can occur in various settings such as intra- and inter-organizations, social sector, government, international development, the Web, and associations (Wenger 2011).

CoPs have evolved from single-organization communities (Ben Saad 2020) to multi-communities and organizations. In a multi-community arrangement, an individual is formally attached to one organization while informally belonging to multiple CoPs (Ishiyama 2016; Omidvar and Kislov 2014). This arrangement affects the landscapes of practice and the formation of identity (Omidvar and Kislov 2014).

Establishing CoPs has several benefits. They have been identified as important entities in advancing theory and practice (Amidon 1998), and blurring the line between theory and technology (Su et al. 2012). For academics, their participation in such communities can positively impact their sense of belonging and competence in the knowledge domain of choice (Brown and Peck 2018). Due to the immense positive contributions of CoPs to an organization, some have chosen to mandate their establishment (Ivcovici et al. 2021). The common forms of CoP are physical, virtual, and hybrid.

Physical or virtual CoPs have been identified as important facilitators of knowledge sharing (KS) (Gammelgaard 2010; Ogbamichael and Warden 2018). Even in the public sector, the establishment of CoPs can positively affect KS, which can eventually lead to improvements in service delivery to the public (Jørgensen et al. 2021). Organizations can connect these clusters of knowledge and expertise, enabling them to collaborate and address similar problems. Consequently, a support network where new ideas and opportunities are shared will emerge, while standards and best practices are established. This is the precursor to building knowledge-based organizations (Mitran et al. 2009).

The tax community presents an interesting and dynamic landscape of a CoP, similar to the one presented by the epidemiologists' community. Despite the similarities, there is one major difference: the activity of a CoP for epidemiologists' peaks during an epidemic or pandemic crisis like COVID-19 while the CoP for tax professionals changes annually when a new budget or law that impacts the way individuals and businesses are taxed is introduced. These announcements must be translated into a knowledge form that can be easily understood by the public aiming to reduce tax avoidance and/or tax evasion (Al-Rahamneh and Bidin 2022; Khalid et al. 2021). Therefore, various parties (e.g., the Ministry of Finance or Inland Revenue Board of Malaysia, tax associations, academicians, and other professionals) are involved in the CoPs for tax professionals and are responsible for disseminating new knowledge to their stakeholders. For example, academics are tasked with sharing knowledge with their students, tax agents with their clients, and various professional bodies with their members, as well as the accounting fraternity.

Unfortunately, little is known about how taxation communities collaborate on their knowledge. Most empirical tax research has employed quantitative methods and focused on the operational and technical issues of taxation such as the framework for cryptocurrency taxation (Caliskan 2022), the impact of the effective tax rate towards capital structure (Ali



et al. 2022), tax avoidance (Dyreng et al. 2019; Armstrong et al. 2015; Utomo et al. 2015), methods of tax accounting (Bergner and Heckemeyer 2017), and tax compliance (Hassan et al. 2022; Wahab and Bakar 2021). Studies that focus on collaborating on knowledge among the tax community, especially its processes, are limited to the knowledge management (KM) domain, for example, sharing and communicating knowledge (Setyorini et al. 2019; Hasseldine et al. 2012; Okoh et al. 2021).

This lack of knowledge represents a missed opportunity. Understanding the tax community's KC process could enable better comprehension of KM dynamics. For example, it could enhance our understanding of how actors within multiple agencies work together to improve the public's knowledge of a specific domain. Additionally, it can broaden the research perspective by explaining how and why relationships among actors exist. The dynamism involved in the tax community has motivated this study to use it as its research context. Scholars argue that CoPs, especially online ones, may not be successful without members who are willing to spend time and effort to practice KC effectively for the community (Faraj et al. 2011; Rheingold 2000).

To the best of our knowledge, this is the first study to focus on KC among tax professionals within Malaysian accounting associations. We aim to examine KC and interaction among tax professionals through the lens of CoPs. The main questions that we seek to answer are presented below.

RQ 1: How do tax professionals collaborate and interact with each other to attain the highest standard of technical and professional competency in tax knowledge and practice?

RQ 2: What mechanisms do professional associations employ to facilitate KC among community members?

## 2. Literature Review

### 2.1. Knowledge Collaboration Processes in Communities

According to Cheng and Chang (2020), KC research is still in its initial development stage. Nevertheless, various views have been identified and classified into three groups—process, mobility, and synergy theories—based on the existing literature. First, the theory of process views KC as a constantly progressing and emphasises the process of achieving innovation in knowledge, which normally occurs in the form of a two-way dynamic flow (Zou and Wang 2016). Second, the theory of mobility emphasises collaborative knowledge activity from the viewpoint of mobility and flexibility. It regards KC as a mobile and innovative activity that involves different individuals in a clustered environment (Dan 2009) and exploits knowledge resources (Salavisa et al. 2012). Third, the theory of synergy is used as a frame of reference for surplus value and value-added effects generated by the synergistic action of KC (Yue and Xin 2012). Of the three theories, this study focuses on the theory of process, to contribute to the literature on KC in the taxation field.

The literature on KC categorises various processes within KC frontiers. Scholars broadly define KC as the sharing, transfer, accumulation, transformation, and co-creation of knowledge (Faraj et al. 2011; Liu et al. 2011); transfer, translation, and transformation of knowledge (Randhawa et al. 2017); and sharing, creation, and collaboration of knowledge (Liu et al. 2020). These processes occur extensively, especially in communities where members progressively communicate and interact with each other. Collaborative factor is one of many non-technical factors that has been reported help to facilitate sharing of knowledge within a community (Mohamad Judi et al. 2018). Therefore, a growing body of literature examines the communication and interaction among community members, enabling KC.

In a broader context, KC studies have investigated various communities: online, virtual, hybrid-virtual (from different industries), and geographically dispersed collaborations beyond organisational boundaries (Cramton 2001; Jarvenpaa and Majchrzak 2010; Faraj et al. 2011; Grabher and Ibert 2014). Nevertheless, a more recent literature review suggests that many studies on KC have been conducted in online communities. Online communities include open-source software (OSS) communities (Kakimoto et al. 2006); firm-hosted,

firm-related, and independent communities (Grabher and Ibert 2014); Usenet newsgroups (Faraj et al. 2015); Wikipedia (Mansour et al. 2011; Park and Park 2016; Wang et al. 2019); citizen sourcing by Nexus (Randhawa et al. 2017); and online encyclopaedias (Liu et al. 2020). The findings from these studies demonstrate that KC has a significant impact on communication and interaction among members, and ultimately increases collaboration efficiency.

Some of these studies draw on the CoP perspective. For example, Randhawa et al. (2017) examined KC between organisations and online-user communities by drawing on CoP perspectives on knowledge. This study employed CoP based on its perspective that views knowledge as embedded and localised in practice when it is invested by the members (Lave and Wenger 1991; Brown and Duguid 1991). Similarly, Mansour et al. (2011) used CoPs as evidence to study interactions among members within a community. Activities performed by the members, such as sharing, accumulating, and transferring knowledge, reflect the KC process.

Combining the CoP perspective with community-driven activities and products has the potential to enhance the outcomes of their actions. The combination will lead to the search for and identification of expertise in the community. Once this expertise is found, the community will be able to streamline their actions and reduce redundant efforts. Eventually, the synergy achieved will benefit the community itself and the larger public through better planned programs such as seminars, talks, training, and publications.

In the following section, we discuss the CoP in greater detail.

### 2.2. Factor and Process Studies on CoPs

Many studies have investigated the factors affecting the effectiveness of CoPs. These studies are generally divided into two streams: one investigating the characteristics of members in a CoP and the other investigating the role of the top management team. In terms of COP members' characteristics, studies have found some factors could determine the effectiveness of the communities. For example, Hafeez et al. (2019) found that the members' intensity of engagement was a strong indicator of the strength of KS. In another study, Iverson and McPhee (2008) found that the elements of mutual engagement, negotiation of a joint enterprise, and a shared repertoire affect the communicative nature of knowing, which eventually affects the CoP's KS effectiveness.

The top management team's handling ability is an important factor in the effectiveness of a CoP. For formally managed CoPs, a supportive environment must be created for communities to prosper (Jeon et al. 2011a, 2011b). They must also be aware of possible Machiavellian participants who might negatively affect the communities' effectiveness (Schofield et al. 2018). Thus, the right incentive mechanisms need to be put in place to ensure that the right behaviour is inculcated in the CoPs (Li and Jhang-Li 2010). Contrastingly, a study suggests that there is no evidence to indicate that rewards can enhance the motivation to share knowledge (Zamri and Ithiran 2021).

For KM and KS to occur successfully within and between firms, they need to be complemented with appropriate mechanisms or process models (de Sousa and de Souza 2019). KS involves the dynamic process of forming, renewing, and reshaping collective and reciprocal relationships (Marsick et al. 2014). These relationships enable the authentic experiences and needs of communities to be captured through iterative and community-driven processes and products (Majeski and Schefkind 2021). A good CoP model can enhance understanding of organizational culture, business strategy, and performance measurement (Chu et al. 2012).

Learning is an important process in a CoP. It is a social process in which a community engages in common work practices, creates knowledge, and shares ways of knowing (Carter and Adkins 2017). This learning process enables tacit and implicit knowledge to be transferred from the experts to the novices (Choi 2006). It is also known as "knowledge brokering," wherein an actor acts as a gatekeeper or boundary spanner by providing new and external information to other organisation members (Ishiyama 2016). With the

assistance of a knowledge-sharing-friendly culture, this process can be accelerated as trust is established between the experts and novices (Gammelgaard 2010). Additionally, a change in personnel can affect the knowledge-diffusion process (Huang et al. 2007).

An often-neglected component of CoPs is identity formation, which occurs during knowledge construction, meaning making, and interaction (Evnitskaya and Morton 2011; Ivcovici et al. 2021). Its success depends on the understanding of the connecting mechanisms between colleagues in a professional setting, such as organisational control, organisational opportunity, social networks, and non-person-centred approaches (Wanberg et al. 2017). Continuous failure to appreciate this process can lead to unintended participation trajectories between experts and novices (Ivcovici et al. 2021). Consequently, it can also affect the speed at which KC motivations and behaviours occur (Wang et al. 2019).

The appropriate information technology (IT) can support the knowledge-sharing process in a CoP. For example, a knowledge-map management system can be used to assist a community to manage knowledge in a virtual environment (Lin and Hsueh 2006). Curran et al. (2009) found that rural emergency clinicians depended on online discussion boards to acquire relevant health diagnoses and medication. A more comprehensive system can include functions for KS, publishing and forwarding, recommendations, peer clustering, and instant messaging (Wang et al. 2008). However, the success of these systems depends on several factors: a flexible KM strategy, multiple channels for KS, the desire to expand the CoP, and the evolution of IT in tandem with KM strategies (Pan and Leidner 2003).

### 2.3. Tax Practices

Actors in the Malaysian tax system include tax advisors, taxpayers, and tax legislators. The tax legislator in Malaysia is the Inland Revenue Board (IRB), which primarily acts as the knowledge supplier and performs roles ranging from the interpretation of various pieces of tax legislation and administration activities, including acting as initiators for Malaysian tax reforms. Tax legislation is a necessary public policy tool to change or encourage actions or activities, such as investment in research and development-related activities and the timely transfer of information on recent changes in tax legislation. An efficient tax system is characterised by how tax knowledge is managed, and taxpayers become aware of tax legislation and other related information (Hasseldine et al. 2012). Therefore, tax knowledge is an important prerequisite to facilitate effective corporate tax planning for both individuals and corporations. Research has shown that the tax compliance level has always been low and tax knowledge is an important factor that influences tax compliance (Amin et al. 2022). The IRB also ensures compliance and engages in consultation with the relevant stakeholders within the Malaysian taxation ecosystem by putting in place strategies and structures that minimise non-compliance with tax legislation among the taxpayers. Taxpayers are broadly categorized into four types based on the extent to which they comply with their obligations: (a) registration in the system; (b) timely filing of requisite taxation information; (c) reporting of complete and accurate information; and (d) settling taxes payable on time.

Tax knowledge is the level of awareness or consciousness of taxpayers about tax legislation, including the process of taxation and other tax-related information (Hantono 2021). Tax knowledge has been conceptualised in the literature as general fiscal knowledge (Groenland and van Veldhoven 1983) and transformed into a more refined as the taxpayers' knowledge of tax matters (Eriksen and Fallan 1996) as well as taxpayers' knowledge and competencies in filing their tax returns (Kasipillai 1997) including the knowledge of tax rules and financial knowledge to allow the calculation of economic consequences (Palil 2005). Bornman and Ramutumbu (2019) further identified that there are three elements of tax knowledge, namely, general (relates to a need to have a fiscal awareness), procedural (understanding tax-compliance procedures) and legal tax knowledge (knowledge pertaining to a need to understand regulations). Tax knowledge is crucial as it can directly influence the tax compliance, in that limited tax knowledge might set the taxpayers back from complying with tax regulations (Ramutumbu 2016).

### 3. Research Methodology

This study employs a qualitative approach. The main aim of our study is to gain insight into how professionals in the taxation field collaborate on knowledge and interact with each other in a CoP. Based on extensive discussion of the KC process and CoPs in the previous section, we propose the following initial research model (Figure 1) to examine KC and interactions among Malaysian tax professionals. Realizing that practices are embedded within the community and domain; we identified the broad KC processes found in the literature.

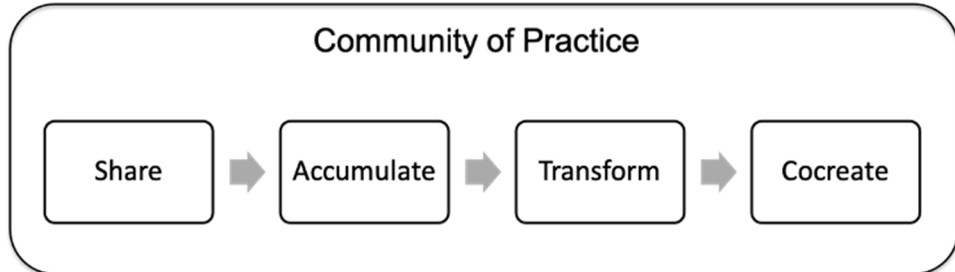

**Figure 1.** Initial research model.

Data were collected from semi-structured interviews, field notes, and archives (Table 1). We conducted semi-structured in-depth interviews, each lasting an hour on average. In-depth semi-structured interviews were used to collect data from the respondents. The interview methods enabled this study to obtain a better understanding of the detailed perspective of the knowledge-sharing phenomena amongst tax practitioners. The participants were 29 tax professionals and included academics, tax advisers, and employees from the IRBM. The tax advisers comprised companies drawn from various settings to increase the variety of collaborations and interactions. They can be classified into three main categories: those who prepare tax returns for the company with which they work (Category 1); those who work with accounting firms, including the Big Four, and prepare tax returns for its clients (Category 2); and those who own a consultancy firm and prepare tax returns for clients, including private companies, sole proprietorships, partnerships, and individuals (Category 3).

**Table 1.** Data sources.

| Source | Description |
|---|---|
| | Academicians–14 interviews |
| | IRBM employees–3 interviews |
| Interview | Tax advisers–12 interviews |
| | Category 1 (3 interviews) |
| | Category 2 (6 interviews) |
| | Category 3 (3 interviews) |
| Field notes | Observations during the interview. |
| Archival data | Websites, media releases, books, and scholarly journals |

The informants were affiliated with at least one of the following professional associations: the Malaysian Institute of Accountants (MIA), Malaysian Institute of Certified Public Accountants (MICPA), Chartered Tax Institute of Malaysia (CTIM), CPA Australia, Association of Chartered Certified Accountants (ACCA), Institute of Chartered Accountants in England and Wales (ICAEW), and Chartered Accountants and Chartered Institute of Management Accountants (CIMA). The number of members in each association is high; for example, MIA has more than 35,500 members working across all Malaysian industries and states. These associations conduct numerous activities to benefit their members in

both offline and online modes. Collaboration and interactions among members inter- or intra-association can be considered a CoP.

The informants were asked open-ended and semi-structured questions. The questions are categorized in eight focus areas, namely, share, transfer, accumulate, transform, cocreate, technology, and community of practice (see Appendix A). These focus areas were created from concepts drawn from the literature. The questions are related to how the informants practice knowledge collaboration in the taxation community and how technology facilitates the process of sharing, transferring, accumulating, transforming, and cocreating tax knowledge. We also probed further when we wanted to learn more about informants' experiences and perceptions to attain a richer understanding.

The interviews were recorded, and notes were written during and after the interview. Next, the recorded interviews were transcribed verbatim in preparation for data analysis. This study uses deductive analysis for analysing the interview transcripts (Azungah 2018). Based on the predetermined focus area in the interview questions, we mapped connections in the data to those specific categories. During the analysis process, codes are grouped into clusters around similar and interrelated ideas and concepts. Themes are then articulated and developed through comparison between and within as obtained from the respondents in this study.

## 4. Results

In this section, we further discuss the community members' practices and map them with the KC process.

### 4.1. Malaysian Accounting Associations as CoPs

The informants' engagement in their domain of interest with other practitioners in an association reflects a CoP. The CoP concept can be described by discussing three components: a knowledge domain, a community, and a shared practice, as represented in Figure 2.

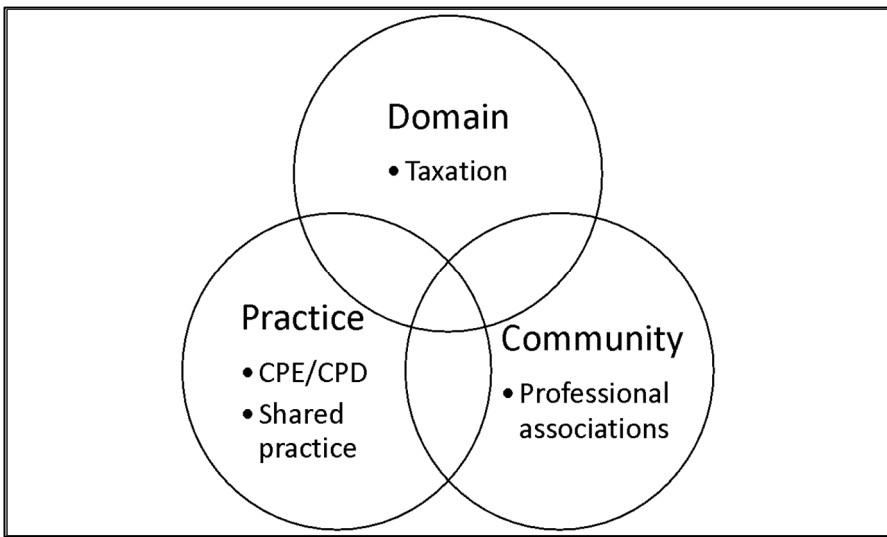

**Figure 2.** Community of practice in the taxation field.

### 4.1.1. Tax Knowledge as the Knowledge Domain

The results indicate that the informants are a group of people who have an identity defined by their shared domain of interest, namely taxation practice. They engage in collective learning and joint activities to develop a shared repertoire of resources as well as expanding their expertise and knowledge. For example, most accounting associations in Malaysia are actively organizing seminars and conferences that act as a platform for their members to interact with each other through various activities that results in the

production of knowledge. In fact, they have organized more online activities during the global pandemic of COVID-19 and the participation rate was high. According to one of the informants who attended the online activities, "It gives me a greater flexibility to attend online conference as compared to attending the conference physically. I do not have to rearrange my work schedule and I can attend more conferences than I normally do."

Most of the informants agreed that to attain the highest standard of technical and professional competency and stay relevant in the field, they must continuously update their tax knowledge. We found several types of tax-related information that can be associated with tax knowledge that are deemed important for professionals in the field such as public rulings, planning, rates, legislation, relief, exemptions, deductions, taxes payable, incentives, returns, rebates, assessments, penalties, risks, treatment, withholding, borne, computation, policies, liabilities, savings, and services. These terms appeared more frequently in the interview transcripts were deemed prominent. We then identify a few subdomains of tax knowledge from the derived words. The sub-domains are legislation, planning, execution, and compliance. Knowledge of these subdomains is deemed crucial for informants to perform their role as tax advisers.

### 4.1.2. A Professional Association as a Community

The main purpose of these associations is to provide professional development to their members to support economic growth and nation-building by updating members on technical information and the latest laws, policies, and guidelines. Furthermore, they assist members to establish their practices. The associations have helped connect and link members from various backgrounds, such as tax lawyers, tax agents, tax practices, academicians, and students. The linkages include collaboration between universities and the industry, such as encouraging placement programs for lecturers in industries, industry players teaching at universities, and applied research. In addition, KC also occurs among practitioners and organizations, where they discuss clients' cases in a working group and collectively identify a common problem-solving approach.

### 4.1.3. Members in the CoP

This study includes members from various backgrounds, including academics and practitioners in the taxation field. We also included employees from the IRBM to provide insights into the Malaysian tax system and the entities involved in it. We then classified these professionals into three broad groups: academics, IRBM employees, and tax advisers. Our findings demonstrate that every member in the CoP interacts and engages with each other to share expertise and experiences through joint engagement in inquiry. Most of the informants felt that it is important to build networks and connect to ensure growth in their professional development and to further improve the tax framework to ensure a better fiscal position and economic growth. Additionally, such engagement offers access to a shared repertoire of resources developed by the community, especially in terms of operational and technical knowledge. From the top-down view, the IRBM supplies tax-related knowledge to tax professionals, especially on tax-legislation matters. However, at various stages, the IRBM's roles may change; for example, in some settings, academicians and tax advisers may act as knowledge suppliers to the IRBM through findings from academic research and/or taxpayers' feedback that tax advisers have dealt with. It is evident that members of the CoP collaborate and interact in many ways, especially through shared practice. The supposed interactions of supplying and retrieving tax knowledge among the three members in our study are presented in Figure 3.

Through shared practices, informants learn from more experienced members. They gradually increase their knowledge and level of participation in the domain until they are experienced enough to impart knowledge to novices. There are many programs and activities organized by the associations to support their members' professional growth through training and learning. The main programs are Continuing Professional Education (CPE) and Continuing Professional Development (CPD). Over the years, the associations

have organized numerous activities (as exhibited in Table 2) such as seminars, webinars, case studies, conferences, in-house training, surveillance and enforcement, commemorative lectures, publications, dialogues, workshops, and circulars. These activities have become a source of knowledge for the members and have equipped them with education, training, technical support, and advocacy.

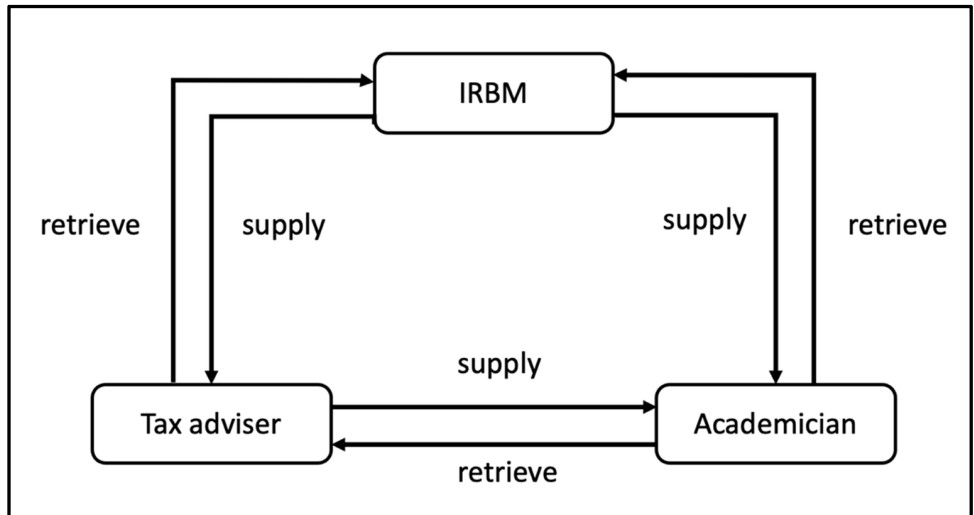

**Figure 3.** Tax-knowledge flows among entities.

**Table 2.** List of knowledge sources and activities.

| Activity | Example of Activities | Association(s) |
| --- | --- | --- |
| Webinar | Tax withholding, FY 2020 transfer-pricing report, payroll-tax computation. | MIA, CPA |
| | Tax audit, tax appeal procedures, interest expense. | ACCA, CTIM, IRBM |
| | Tax-aligned mergers and acquisitions, tax litigation, tax audits, and investigations. | CTIM |
| | Implication for business restructuring, public rulings in 2020 and 2021, employee tax reporting. | CPA, IRBM |
| | Malaysia Budget 2021 | Ernst and Young |
| | Talent transfer to Malaysia, transfer pricing, and payment to individuals. | KPMG |
| | i. BDO Tax Webinar: Updates on Service Tax ii. BDO Tax Budget Seminar 2019 | BDO |
| Case Study | Deferred taxation in complex transactions and events | MIA |
| eLearning | Ethics in tax | MIA |
| | Expert hour, transfer-pricing updates, tax awareness | Crowe Horwath |

**Table 2.** *Cont.*

| Activity | Example of Activities | Association(s) |
|---|---|---|
| Virtual Conference/Conference | i. Data Intelligence and Analytics 2.0 Conference for Public Sector<br>ii. Islamic Finance Conference 2021 | MIA |
| | MIA Malaysian Tax Conference 2021 | ACCA, MIA, CIMA, MICPA |
| | i. Tax Max<br>ii. National Tax Conference 2019<br>iii. Transfer Pricing Conference 2019 | Deloitte, IRBM, CPA, MIA |
| In House Training | Tailor-made programs | MIA |
| Technical Resources | Budget booklets, cases, public rulings, budget and financial bills, service taxes and dialogues with IRB and Customs: technical, operational, desire working group, workshops. | MIA |
| | Taxation in Malaysia–by country resources, including tax treaties, news and developments, rates and guides, library collections. | ICAEW |
| | Public Rulings | IRBM |
| Surveillance and Enforcement | Conducts practice and financial statements, review, continuing professional education. compliance audit, investigation, and disciplinary proceedings | MIA |
| | i. Review and set guidelines in Malaysia–International Accounting Standards (IAS) and International Standards on Auditing (ISA) Develop accounting and reporting standards for specialised industries (e.g., insurance, banking, and financial services) | MICPA |
| Member's Handbook | Provides technical and professional standards. | MICPA |
| | Factsheets and guidelines | ACCA |
| Commemorative Lecture | 63rd Anniversary Commemorative Lecture in July 2021 | MICPA |
| Publications | Approved Accounting Standards Malaysian Financial Reporting Standards (MFRS) and Private Entities Reporting Standard (MPERS) | MICPA |
| | Technical updates, tax and investment review, budget commentary and tax information | MICPA |
| | Books and magazines and the Malaysian Accountant Journal | MICPA, CTIM, and MIA |
| | The Tax Guardian journal, published quarterly | CTIM |
| | Articles such as "Transfer pricing update: Malaysia tightens transfer pricing compliance requirements, and Budget 2022–Part I" (Tax Espresso Special Edition) | BDO Deloitte |
| | TaXavvy | Pricewaterhouse Coopers |

**Table 2.** *Cont.*

| Activity | Example of Activities | Association(s) |
|---|---|---|
| e-Circular/Circular | Loans Guarantee (Bodies Corporate) (Remission of Tax and Stamp Duty) (No. 2) Order 2021 [P.U. (A) 322/2021] | CTIM |
| | Monthly newsletter via email to taxpayers | LHDNM |
| | Tax Alerts subscription | Ernst and Young |
| | Tax Espresso subscription | Deloitte |
| | KPMG's insights subscription | KPMG |
| | Newsletter–Tax | Crowe Horwath |
| Blog/Discussion board | Tax Whiz Club (LinkedIn group) | KPMG |
| | Our Perspective on Tax | Pricewaterhouse Coopers |

### 4.1.4. Informal Communication and Interaction among Members

The communication and interaction among members are not limited to formal activities organised by associations or organisations. It was found that tax professionals engage in informal activities with those they are close to, both within and outside their organisations. They believe that such informal sessions increase the possibility of communication and interaction. Thus, knowledge exchange is easier and it is more convenient to obtain insights from a particular person. These informal conversations and exchanges of ideas could happen over lunches, drinks, dinners, post meetings, and communication through instant messaging or phone calls. One of our informants who holds a tax agent position says, "We share stories about concrete cases over a cup of coffee. It is casual and spontaneous but will add value to our job." Examples of informal knowledge-sharing activities are stories, tips, news, information, and pointers to resources. Such activities are especially useful for junior members. One of the junior tax advisors said, "As junior staff, I often seek help from my colleagues, especially senior staff. They are willingly to lend a hand and give their opinion. With their help, I can perform my task better and with confidence." A majority of the informants note that informal learning helps improve their knowledge base and employ new techniques. A learner who sees informal sharing is deemed necessary for an individual who is interested to explore and learn. These circumstances are reflected in the following observations:

> . . . *clients provide information and documentation prior to the preparation of their tax returns; they might not like the result . . . Then, they return with more documentation and request a re-run . . . It will become more complicated, and it requires me to discuss with my peers or seniors and seek their opinion at any time during the day . . . This is because we need to be flexible while remaining tax compliant.* (Tax consultant)

> *We do not have many lecturers who specialise in taxation in a faculty. Normally, there's only one or two lecturers teaching the subject . . . The syllabus, especially public ruling, changes on a yearly basis . . . If I can't interpret the published guide well, I will refer to my industrial contacts . . . Better still to get in touch with employees from IRBM . . . Usually, they can provide ideas about particular tax laws, policies and procedures.* (Lecturer)

### 4.2. Knowledge Collaboration Process

Our results indicate that KC occurs in the CoPs for tax professionals and involves iterative knowledge exchanges between organizations, associations, and community members. For example, KC among taxation professionals occurs when they share knowledge, accumulate it from others, transform it from a tacit to explicit mode, or vice versa, and co-create it by working with others. In this community, KC occurs with the support of associations as

they become the platform for members to interact with each other, learn from, share goals and passion, and continue their professional development. The collaboration is supported by shared practices, such as public discussions, the exchange of expertise and experience, sharing stories and information, joint engagement in inquiry, the development of rules and regulations, providing feedback on newly published tax laws, and identifying a common problem-solving approach and research. Four main mechanisms of KC were found in this study: sharing, accumulation, transformation, and co-creation. Table 3 presents the description and list of activities for each collaboration mechanism.

**Table 3.** List of activities.

| Process | Description | Activities |
|---|---|---|
| Share | Associations to members: Knowledge, insight, and technical expertise. | Training<br>Webinars<br>Newsletters<br>Blog<br>Social media<br>Forum |
| | Members to associations: Compilation of clients' feedback and cases relevant to public rulings implementation. | Case studies<br>Meeting<br>Roundtable<br>Forum |
| | Member to member: Operational and technical tax knowledge. | Meeting<br>Roundtable<br>Coffee talk<br>Stories |
| Accumulate | Associations: qualifications and professional developments. | Continuing Professional Education (CPE)<br>Continuing Professional Development (CPD)<br>Engagement for industry professionals<br>Research |
| Transform | Associations: explicit knowledge. | Newsletter<br>Documentation<br>Articles |
| | Members: convert explicit knowledge to tacit knowledge and vice versa. | Write books and articles<br>Talk<br>Stories<br>Research |
| Cocreate | Members engage in a design or problem-solving process to produce a mutually valued outcome. | Case studies<br>Joint engagement in inquiry |

### 4.3. Summary of Knowledge Collaboration and Interaction among Members

Table 4 summarizes the common collaboration activities among the CoP members. It is apparent that the three main members of the Malaysian taxation domain collaborate and interact with each other in shared practices to pursue their interests in the domain.

This collaboration offers numerous benefits not only to members within the CoP but also to society. Undoubtedly, KC among members could help foster and advance the practice of taxation in all its aspects and maintain high standards of practice and professional conduct. In addition, collaboration could benefit society by increasing public awareness. Taxpayers, both corporate and private, will know and understand how to implement tax regulations properly and eventually fulfil their tax obligations. Apart from the IRBM, companies and universities are also actively engaged with corporate social responsibility (CSR) programs.

Moreover, this collaboration could benefit future talent in the field of taxation. Players in the industry and universities could work together to provide relevant and meaningful activities for university students. These mutually beneficial partnerships serve both companies and universities, where companies get access to a network of faculty and emerging talent and academics have more resources for research and students such as access to real-world experience and data. Such activities allow students to prepare themselves for the workplace and develop practical skills. Figure 4 summarizes the information sources, KC processes, and knowledge presentation to the audience.

**Table 4.** Common collaboration activities among the entities.

|  | IRBM | Academician | Tax Adviser |
|---|---|---|---|
| **IRBM** | • Intra-organisational discussion<br>• Problem-solving<br>• Regulation | • Scholarly activity | • Share case studies<br>• Provide feedback from taxpayers<br>• Give inputs on legislation matters |
| **Academician** |  | • Scholarly activity<br>• Teaching and learning | • Case studies<br>• Guest lectures<br>• Public discussions |
| **Tax Adviser** |  |  | • Share stories and experiences<br>• Joint engagement in inquiries<br>• Relationship-oriented private discussions |

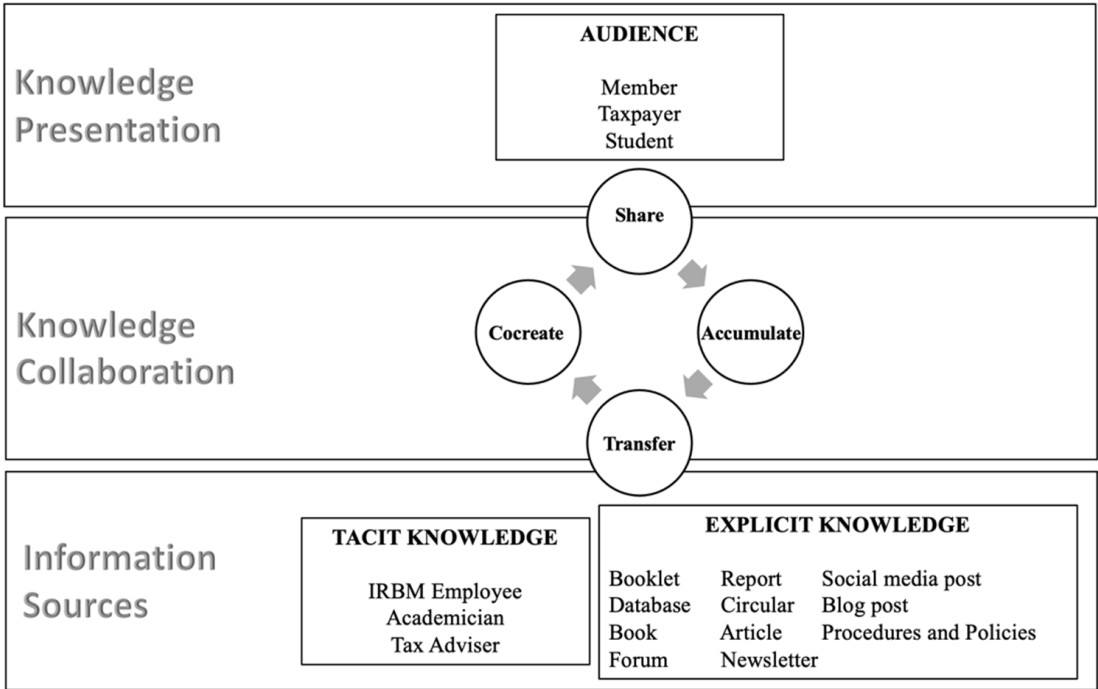

**Figure 4.** Three-layer process.

The community's collaborative behaviour is assisted by collaboration tools, mainly Web 2.0 technologies. Given that the pandemic started in early 2020, most activities were conducted online. Virtual conferences, webinars, online training, meetings, and virtual events were hosted using collaboration tools that focused on enabling communication and interaction among members, including Zoom, Google Meet, and Microsoft Teams. In addition to these tools, informants claimed that they also used instant messaging apps such as WhatsApp, Telegram, Skype, and Google Hangouts. Social media infographics were used to present information visually to the target audience. For instance, the IRBM and Big Four actively share visual information about tax relief on their social media platforms to convey messages briefly. This was especially useful when the Malaysian government announced special relief for businesses and individuals that had been severely impacted by the pandemic.

**5. Conclusions**

In this study, we investigated how tax professionals collaborate and interact with each other in a CoP setting and the mechanisms used to facilitate KC among members within the community. The findings demonstrated that Malaysian accounting professional associations reflected a CoP, and KC occurred among its members. KC among Malaysian taxation professionals occurs when they share, accumulate, transform, and co-create knowledge by working with others. In contrast to prior empirical tax research, this study attempted to examine and uncover KC processes among tax professionals to attain the highest standard of technical and professional competency in tax knowledge and practice.

Theoretically, these findings complement and expand on the previous research on CoPs, KM, KC, tax knowledge, and tax systems and practices. Our study broadens the research perspective and enriches the understanding of the dynamic KC mechanism in the CoPs for tax professionals by integrating the KM and CoP literature, as KC is an advanced stage of KM (Cheng and Chang 2020). Furthermore, new research findings obtained from this study could help enrich and develop theoretical research in this area, as it is in its initial stage.

The insights gained from this study offer several practical implications. First, the taxation community must constantly innovate collaboration mechanisms and attract more participants in collaborative creation to promote seamless collaboration. Second, we found that both formal and informal communication are important for successful KC in the CoPs. Thus, the community could also investigate how communication should be enhanced by identifying barriers that hinder members from sharing their knowledge and exchanging opinions, which is unfavourable to achieving the desired goals. Third, we identified the lack of a central repository for tax knowledge. Related information and knowledge are scattered and hosted by various institutions, organisations, and associations. Therefore, community administrators must explore better strategies to implement a central repository of knowledge to systematically acquire, organise, and categorise knowledge-based information accumulated or generated by members within the community. This repository is important to support the learning cycle.

This study is limited to KC in taxation associations. Future research can confirm whether our findings apply to other communities of practice, including online, virtual, and hybrid-virtual communities from different industries, and geographically dispersed collaboration beyond organisational boundaries. Developing conceptual frameworks, models, and theories is a promising direction for future research.

**Author Contributions:** Conceptualization, N.B. and S.B.; methodology, N.B., S.B. and Z.Z.; validation, Z.Z.; formal analysis, N.B.; investigation, N.B., S.B. and Z.Z; resources, N.B. and Z.Z.; data curation, N.B.; writing—original draft preparation, N.B.; writing—review and editing, S.B.; visualization, N.B.; supervision, S.B. and Z.Z.; project administration, N.B.; funding acquisition, N.B., S.B. and Z.Z. All authors have read and agreed to the published version of the manuscript.

**Funding:** This work was supported by the University of Malaya under the Faculty Research Grant [GPF004I-2019].

**Informed Consent Statement:** Informed consent was obtained from all subjects involved in the study.

**Data Availability Statement:** The data presented in this study are available on request from the corresponding author. The data are not publicly available due to privacy.

**Conflicts of Interest:** The authors declare no conflict of interest.

## Appendix A. Interview Guide

| Area of Interest | Questions |
|---|---|
| Share | 1. What is your opinion about the importance of sharing knowledge in the taxation field?<br>2. Do you believe that sharing one's knowledge and experience may help continual learning and enhanced professional development in this field?<br>3. What are the types of knowledge or information that you commonly share with others?<br>4. Based on your experience, what are the key success factors for an effective knowledge sharing to happen in this field?<br>5. Based on your experience, what are the factors that may limit participation in the knowledge-sharing activity?<br>6. Could you please state the common technological devices, apps, or systems that assist you in sharing knowledge? |
| Transfer | 1. How do you normally transfer your knowledge to others within and across firms?<br>2. Could you please describe some of the situations that require you to transfer knowledge to others?<br>3. How would you describe your motivation to transfer/teach knowledge?<br>4. How would you describe others' motivation to receive/learn the knowledge?<br>5. To what extent is the knowledge-transfer activity among the players in the taxation field meeting expectations? |
| Accumulate | 1. How do you normally access expertise in your field?<br>2. How easy or difficult is it to access expertise in your field?<br>3. Could you please describe other ways, apart from accessing expertise, to gain and accumulate your knowledge in this field?<br>4. How can this be improved? |
| Transform | 1. In what way do you normally transform your personal knowledge into other forms such as articles, books?<br>2. What are your views about transforming one's personal knowledge into something that can be easily accessed by others?<br>3. What are the things that can hinder knowledge transformation? |
| Cocreate | 1. From the knowledge-sharing activities that we have discussed earlier, do you think those activities are really helpful to create new knowledge in your field?<br>2. And how important is it to create new knowledge in this field?<br>3. If that is so, do you think the current practice is sufficient in ensuring new knowledge to be created? |

| Area of Interest | Questions |
|---|---|
| Technology | 1. What technologies do you normally use to share knowledge and/or information with other players in the field?<br>2. Will the members be able to access and leverage the technology?<br>3. Do you think sharing, transforming, and creating knowledge among the members in this association require more personalized tools or technologies? |
| Community of Practice | 1. Being a member in MIA (or other association) and by interacting with other members, how do you think it fosters learning and information sharing?<br>2. In your opinion, how important is the role of this association in providing opportunities for information exchange among the members?<br>3. Based on your experience, how do the association and its members help you with your personal and professional growth in this field?<br>4. What are the common activities organized by the association to promote knowledge exchange among the members?<br>5. What are your recommendations to further improve and optimize the function of these associations? |

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
