# Peer review of "Knowledge Collaboration among Tax Professionals through the Lens of a Community of Practice"

_jrfm, doi:10.3390/jrfm15100439_

Round 1

Reviewer 1 Report

This is a sound paper well researched and written  Today the ideas of communities of practice among professionals is increasingly important.  It would be good to see similar analysis across the world,.  I would suggest you think a little about your Tables to increase their ease of readability.  Your Figures are good.

Author Response

Thank you for your feedback and suggestions. We have incorporated changes to reflect the suggestions.

Please refer to the attached file for our responses.

Reviewer 2 Report

You should better explain interpretation collected data (methods used for collected data) and connection with the results.

Author Response

(The authors gave the same response as above.)

Reviewer 3 Report

The keywords are chosen representatively and highlight both the article and the journal, under the conditions in which the publication is decided.

The corpus of the paper is nicely built, clearly and correctly developed, according to the scientific rules of writing an article.

It is obviously that there is nothing mentioned about the principles of tax knowledge. More details about the CoP components' role could also be an asset.

It could be useful to know how precisely the community-driven processes and products could get together with a good CoP model in enhancing measurements of different sides of organizational culture and strategic business performances. 

The citation elements imposed by the journal on the left side of the article are not detailed.

The bibliographic sources are, in general, correctly stated and easy to validate. The citations within the text are correct. However, some of them do not contain all required bibliographic elements, such as DOI. For example:

- At point 73. Wenger, E. (2011), "Communities of practice: A brief introduction" - missing Scholars bank University of Oregon http://hdl.handle.net/1794/11736.

- At point 12 Carter T.J. & Adkins, B. (2017) the details given are not sufficient to validate the work.

No work published by the Journal of Risk and Financial Management appears cited in this article. It is also possible to add Amin - 2022, Hantono - 2021 by updating the description of tax knowledge.

The style is clear and expressive. The language is generally comprehensible for any potential reader. However, some minor corrections of language are necessary there where the grammar allows some improvements.

Briefly, the paper raises the interest and curiosity of the reader. It is an interesting topic that challenges the scientific spirit. If the minor issues presented above are considered and solved, the paper can be published; once published it could bring quite a lot of citations.

Author Response

(The authors gave the same response as above.)

Reviewer 4 Report

Review of Knowledge Collaboration among Tax Professionals through the lens of community of practice.

The paper is well constructed and cites numerous studies to support the research idea.  There is nothing wrong with the paper.  The results however, are unsurprising and I am not sure what this study adds our knowledge. 

The paper would be improved by reducing the overcited material on knowledge collaboration and communities of practice and providing more information on the interview questions. The paper states that the “actors in the Malaysian tax system include tax advisers, taxpayers and tax legislators.” But then 14 of the 29 interviews are with academics who are which actor?

The paper would be improved by providing the questions asked of the accountants, instead of just providing selected quotes.  The conclusion states that the authors are interested in applying this methodology to other COP, however it would be more interesting to see if there are groups in which KC and collaboration amongst professionals does not exist.

Author Response

(The authors gave the same response as above.)
